# Subspace Embeddings for the Polynomial Kernel

**Haim Avron**
IBM T.J. Watson Research Center
Yorktown Heights, NY 10598
`haimav@us.ibm.com`

**Huy L. Nguyễn**
Simons Institute, UC Berkeley
Berkeley, CA 94720
`hlnguyen@cs.princeton.edu`

**David P. Woodruff**
IBM Almaden Research Center
San Jose, CA 95120
`dpwoodru@us.ibm.com`

## Abstract

Sketching is a powerful dimensionality reduction tool for accelerating statistical learning algorithms. However, its applicability has been limited to a certain extent since the crucial ingredient, the so-called oblivious subspace embedding, can only be applied to data spaces with an explicit representation as the column span or row span of a matrix, while in many settings learning is done in a high-dimensional space implicitly defined by the data matrix via a kernel transformation. We propose the first *fast* oblivious subspace embeddings that are able to embed a space induced by a non-linear kernel *without* explicitly mapping the data to the high-dimensional space. In particular, we propose an embedding for mappings induced by the polynomial kernel. Using the subspace embeddings, we obtain the fastest known algorithms for computing an implicit low rank approximation of the higher-dimension mapping of the data matrix, and for computing an approximate kernel PCA of the data, as well as doing approximate kernel principal component regression.

## 1 Introduction

Sketching has emerged as a powerful dimensionality reduction technique for accelerating statistical learning techniques such as $\ell_p$-regression, low rank approximation, and principal component analysis (PCA) [12, 5, 14]. For natural settings of parameters, this technique has led to the first asymptotically optimal algorithms for a number of these problems, often providing considerable speedups over exact algorithms. Behind many of these remarkable algorithms is a mathematical apparatus known as an *oblivious subspace embedding* (OSE). An OSE is a *data-independent* random transform which is, with high probability, an approximate isometry over the embedded subspace, i.e. $\|Sx\| = (1 \pm \epsilon)\|x\|$ simultaneously for all $x \in V$ where $S$ is the OSE, $V$ is the embedded subspace and $\|\cdot\|$ is some norm of interest. For the OSE to be useful in applications, it is crucial that applying it to a vector or a collection of vectors (a matrix) can be done faster than the intended downstream use.

So far, all OSEs proposed in the literature are for embedding subspaces that have a representation as the column space or row space of an explicitly provided matrix, or close variants of it that admit a fast multiplication given an explicit representation (e.g. [1]). This is quite unsatisfactory in many statistical learning settings. In many cases the input may be described by a moderately sized $n$-by-$d$ sample-by-feature matrix $A$, but the actual learning is done in a much higher (possibly infinite) dimensional space, by mapping each row of $A$ to an high dimensional feature space. Using the *kernel trick* one can access the high dimensional mapped data points through an inner product space,

and thus avoid computing the mapping explicitly. This enables learning in the high-dimensional space even if explicitly computing the mapping (if at all possible) is prohibitive. In such a setting, computing the explicit mapping just to compute an OSE is usually unreasonable, if not impossible (e.g., if the feature space is infinite-dimensional).

The main motivation for this paper is the following question: *is it possible to design OSEs that operate on the high-dimensional space without explicitly mapping the data to that space?*

We propose the first *fast* oblivious subspace embeddings for spaces induced by a non-linear kernel *without* explicitly mapping the data to the high-dimensional space. In particular, we propose an OSE for mappings induced by the polynomial kernel. We then show that the OSE can be used to obtain faster algorithms for the polynomial kernel. Namely, we obtain faster algorithms for approximate kernel PCA and principal component regression.

We now elaborate on these contributions.

**Subspace Embedding for Polynomial Kernel Maps.** Let $k(x, y) = (\langle x, y \rangle + c)^q$ for some constant $c \geq 0$ and positive integer $q$. This is the degree $q$ polynomial kernel function. Without loss of generality we assume that $c = 0$ since a non-zero $c$ can be handled by adding a coordinate of value $\sqrt{c}$ to all of the data points. Let $\phi(x)$ denote the function that maps a $d$-dimensional vector $x$ to the $d^q$-dimensional vector formed by taking the product of all subsets of $q$ coordinates of $x$, i.e. $\phi(v) = v \otimes \ldots \otimes v$ (doing $\otimes$ $q$ times), and let $\phi(A)$ denote the application of $\phi$ to the rows of $A$. $\phi$ is the map that corresponds to the polynomial kernel, that is $k(x, y) = \langle \phi(x), \phi(y) \rangle$, so learning with the data matrix $A$ and the polynomial kernel corresponds to using $\phi(A)$ instead of $A$ in a method that uses linear modeling.

We describe a distribution over $d^q \times O(3^q n^2 / \epsilon^2)$ sketching matrices $S$ so that the mapping $\phi(A) \cdot S$ can be computed in $O(\mathtt{nnz}(A)q) + \mathrm{poly}(3^q n / \epsilon)$ time, where $\mathtt{nnz}(A)$ denotes the number of non-zero entries of $A$. We show that with constant probability arbitrarily close to 1, simultaneously for all $n$-dimensional vectors $z$, $\|z \cdot \phi(A) \cdot S\|_2 = (1 \pm \epsilon) \|z \cdot \phi(A)\|_2$, that is, the entire row-space of $\phi(A)$ is approximately preserved. Additionally, the distribution does not depend on $A$, so it defines an OSE.

It is important to note that while the literature has proposed transformations for non-linear kernels that generate an approximate isometry (e.g. Kernel PCA), or methods that are data independent (like the Random Fourier Features [17]), no method previously had both conditions, and thus they do not constitute an OSE. These conditions are crucial for the algorithmic applications we propose (which we discuss next).

**Applications: Approximate Kernel PCA, PCR.** We say an $n \times k$ matrix $V$ with orthonormal columns spans a rank-$k$ $(1 + \epsilon)$-approximation of an $n \times d$ matrix $A$ if $\|A - VV^T A\|_F \leq (1 + \epsilon) \|A - A_k\|_F$, where $\|A\|_F$ is the Frobenius norm of $A$ and $A_k = \arg \min_{X \text{ of rank } k} \|A - X\|_F$. We state our results for constant $q$.

In $O(\mathtt{nnz}(A)) + n \cdot \mathrm{poly}(k / \epsilon)$ time an $n \times k$ matrix $V$ with orthonormal columns can be computed, for which $\|\phi(A) - VV^T \phi(A)\|_F \leq (1 + \epsilon) \|\phi(A) - [\phi(A)]_k\|_F$, where $[\phi(A)]_k$ denotes the best rank-$k$ approximation to $\phi(A)$. The $k$-dimensional subspace $V$ of $\mathbb{R}^n$ can be thought of as an approximation to the top $k$ left singular vectors of $\phi(A)$. The only alternative algorithm we are aware of, which doesn't take time at least $d^q$, would be to first compute the Gram matrix $\phi(A) \cdot \phi(A)^T$ in $O(n^2 d)$ time, and then compute a low rank approximation, which, while this computation can also exploit sparsity in $A$, is much slower since the Gram matrix is often dense and requires $\Omega(n^2)$ time just to write down.

Given $V$, we show how to obtain a low rank approximation to $\phi(A)$. Our algorithm computes three matrices $V, U$, and $R$, for which $\|\phi(A) - V \cdot U \cdot \phi(R)\|_F \leq (1 + \epsilon) \|\phi(A) - [\phi(A)]_k\|_F$. This representation is useful, since given a point $y \in \mathbb{R}^d$, we can compute $\phi(R) \cdot \phi(y)$ quickly using the kernel trick. The total time to compute the low rank approximation is $O(\mathtt{nnz}(A)) + (n + d) \cdot \mathrm{poly}(k / \epsilon)$. This is considerably faster than standard kernel PCA which first computes the Gram matrix of $\phi(A)$.

We also show how the subspace $V$ can be used to regularize and speed up various learning algorithms with the polynomial kernel. For example, we can use the subspace $V$ to solve regression problems

of the form $\min_x \|Vx - b\|_2$, an approximate form of principal component regression [8]. This can serve as a form of regularization, which is required as the problem $\min_x \|\phi(A)x - b\|_2$ is usually underdetermined. A popular alternative form of regularization is to use kernel ridge regression, which requires $O(n^2d)$ operations. As $\texttt{nnz}(A) \leq nd$, our method is again faster.

**Our Techniques and Related Work.** Pagh recently introduced the TENSORSKETCH algorithm [14], which combines the earlier COUNTSKETCH of Charikar et al. [3] with the Fast Fourier Transform (FFT) in a clever way. Pagh originally applied TENSORSKETCH for compressing matrix multiplication. Pham and Pagh then showed that TENSORSKETCH can also be used for statistical learning with the polynomial kernel [16].

However, it was unclear whether TENSORSKETCH can be used to approximately preserve entire subspaces of points (and thus can be used as an OSE). Indeed, Pham and Pagh show that a fixed point $v \in \mathbb{R}^d$ has the property that for the TENSORSKETCH sketching matrix $S$, $\|\phi(v) \cdot S\|_2 = (1 \pm \epsilon)\|\phi(v)\|_2$ with constant probability. To obtain a high probability bound using their results, the authors take a median of several independent sketches. Given a high probability bound, one can use a net argument to show that the sketch is correct for all vectors $v$ in an $n$-dimensional subspace of $\mathbb{R}^d$. The median operation results in a non-convex embedding, and it is not clear how to efficiently solve optimization problems in the sketch space with such an embedding. Moreover, since $n$ independent sketches are needed for probability $1 - \exp(-n)$, the running time will be at least $n \cdot \texttt{nnz}(A)$, whereas we seek only $\texttt{nnz}(A)$ time.

Recently, Clarkson and Woodruff [5] showed that COUNTSKETCH can be used to provide a subspace embedding, that is, simultaneously for all $v \in V$, $\|\phi(v) \cdot S\|_2 = (1 \pm \epsilon)\|\phi(v)\|_2$. TENSORSKETCH can be seen as a very restricted form of COUNTSKETCH, where the additional restrictions enable its fast running time on inputs which are tensor products. In particular, the hash functions in TENSORSKETCH are only 3-wise independent. Nelson and Nguyen [13] showed that COUNTSKETCH still provides a subspace embedding if the entries are chosen from a 4-wise independent distribution. We significantly extend their analysis, and in particular show that 3-wise independence suffices for COUNTSKETCH to provide an OSE, and that TENSORSKETCH indeed provides an OSE.

We stress that all previous work on sketching the polynomial kernel suffers from the drawback described above, that is, it provides no provable guarantees for preserving an entire subspace, which is needed, e.g., for low rank approximation. This is true even of the sketching methods for polynomial kernels that do not use TENSORSKETCH [10, 7], as it only provides tail bounds for preserving the norm of a fixed vector, and has the aforementioned problems of extending it to a subspace, i.e., boosting the probability of error to be enough to union bound over net vectors in a subspace would require increasing the running time by a factor equal to the dimension of the subspace.

After we show that TENSORSKETCH is an OSE, we need to show how to use it in applications. An unusual aspect is that for a TENSORSKETCH matrix $S$, we can compute $\phi(A) \cdot S$ very efficiently, as shown by Pagh [14], but computing $S \cdot \phi(A)$ is not known to be efficiently computable, and indeed, for degree-2 polynomial kernels this can be shown to be as hard as general rectangular matrix multiplication. In general, even writing down $S \cdot \phi(A)$ would take a prohibitive $d^q$ amount of time. We thus need to design algorithms which only sketch on one side of $\phi(A)$.

Another line of research related to ours is that on random features maps, pioneered in the seminal paper of Rahimi and Recht [17] and extended by several papers a recent fast variant [11]. The goal in this line of research is to construct randomized feature maps $\Psi(\cdot)$ so that the Euclidean inner product $\langle \Psi(u), \Psi(v) \rangle$ closely approximates the value of $k(u, v)$ where $k$ is the kernel; the mapping $\Psi(\cdot)$ is dependent on the kernel. Theoretical analysis has focused so far on showing that $\langle \Psi(u), \Psi(v) \rangle$ is indeed close to $k(u, v)$. This is also the kind of approach that Pham and Pagh [16] use to analyze TENSORSKETCH. The problem with this kind of analysis is that it is hard to relate it to downstream metrics like generalization error and thus, in a sense, the algorithm remains a heuristic. In contrast, our approach based on OSEs provides a mathematical framework for analyzing the mappings, to reason about their downstream use, and to utilize various tools from numerical linear algebra in conjunction with them, as we show in this paper. We also note that in to contrary to random feature maps, TENSORSKETCH is attuned to taking advantage of possible input sparsity. e.g. Le et al. [11] method requires computing the Walsh-Hadamard transform, whose running time is independent of the sparsity.

## 2 Background: COUNTSKETCH and TENSORSKETCH

We start by describing the COUNTSKETCH transform [3]. Let $m$ be the target dimension. When applied to $d$-dimensional vectors, the transform is specified by a 2-wise independent hash function $h : [d] \rightarrow [m]$ and a 2-wise independent sign function $s : [d] \rightarrow \{-1, +1\}$. When applied to $v$, the value at coordinate $i$ of the output, $i = 1, 2, \ldots, m$ is $\sum_{j|h(j)=i} s(j)v_j$. Note that COUNTSKETCH can be represented as a $m \times d$ matrix in which the $j$-th column contains a single non-zero entry $s(j)$ in the $h(j)$-th row.

We now describe the TENSORSKETCH transform [14]. Suppose we are given a point $v \in \mathbb{R}^d$ and so $\phi(v) \in \mathbb{R}^{d^q}$, and the target dimension is again $m$. The transform is specified using $q$ 3-wise independent hash functions $h_1, \ldots, h_q : [d] \rightarrow [m]$, and $q$ 4-wise independent sign functions $s_1, \ldots, s_q : [d] \rightarrow \{+1, -1\}$. TENSORSKETCH applied to $v$ is then COUNTSKETCH applied to $\phi(v)$ with hash function $H : [d^q] \rightarrow [m]$ and sign function $S : [d^q] \rightarrow \{+1, -1\}$ defined as follows:

$$H(i_1, \ldots, i_q) = h_1(i_1) + h_2(i_2) + \cdots + h_q(i_q) \bmod m,$$

and

$$S(i_1, \ldots, i_q) = s_1(i_1) \cdot s_2(i_1) \cdots s_q(i_q).$$

It is well-known that if $H$ is constructed this way, then it is 3-wise independent [2, 15]. Unlike the work of Pham and Pagh [16], which only used that $H$ was 2-wise independent, our analysis needs this stronger property of $H$.

The TENSORSKETCH transform can be applied to $v$ without computing $\phi(v)$ as follows. First, compute the polynomials

$$p_\ell(x) = \sum_{i=0}^{B-1} x^i \sum_{j|h_\ell(j)=i} v_j \cdot s_\ell(j),$$

for $\ell = 1, 2, \ldots, q$. A calculation [14] shows

$$\prod_{\ell=1}^{q} p_\ell(x) \bmod (x^B - 1) = \sum_{i=0}^{B-1} x^i \sum_{(j_1, \ldots, j_q)|H(j_1, \ldots, j_q)=i} v_{j_1} \cdots v_{j_q} S(j_1, \ldots, j_q),$$

that is, the coefficients of the product of the $q$ polynomials mod $(x^m - 1)$ form the value of TENSORSKETCH($v$). Pagh observed that this product of polynomials can be computed in $O(qm \log m)$ time using the Fast Fourier Transform. As it takes $O(q\,\mathrm{nnz}(v))$ time to form the $q$ polynomials, the overall time to compute TENSORSKETCH($v$) is $O(q(\mathrm{nnz}(v) + m \log m))$.

## 3 TENSORSKETCH is an Oblivious Subspace Embedding

Let $S$ be the $d^q \times m$ matrix such that TENSORSKETCH($v$) is $\phi(v) \cdot S$ for a randomly selected TENSORSKETCH. Notice that $S$ is a random matrix. In the rest of the paper, we refer to such a matrix as a TENSORSKETCH matrix with an appropriate number of columns i.e. the number of hash buckets. We will show that $S$ is an oblivious subspace embedding for subspaces in $\mathbb{R}^{d^q}$ for appropriate values of $m$. Notice that $S$ has exactly one non-zero entry per row. The index of the non-zero in the row $(i_1, \ldots, i_q)$ is $H(i_1, \ldots, i_q) = \sum_{j=1}^{q} h_j(i_j) \bmod m$. Let $\delta_{a,b}$ be the indicator random variable of whether $S_{a,b}$ is non-zero. The sign of the non-zero entry in row $(i_1, \ldots, i_q)$ is $S(i_1, \ldots, i_q) = \prod_{j=1}^{q} s_j(i_j)$. Our main result is that the embedding matrix $S$ of TENSORSKETCH can be used to approximate matrix product and is a subspace embedding (OSE).

**Theorem 1** (Main Theorem). *Let $S$ be the $d^q \times m$ matrix such that TENSORSKETCH($v$) is $\phi(v)S$ for a randomly selected TENSORSKETCH. The matrix $S$ satisfies the following two properties.*

1. *(Approximate Matrix Product:) Let $A$ and $B$ be matrices with $d^q$ rows. For $m \geq (2 + 3^q)/(\epsilon^2 \delta)$, we have*

$$\Pr[\|A^T SS^T B - A^T B\|_F^2 \leq \epsilon^2 \|A\|_F^2 \|B\|_F^2] \geq 1 - \delta$$

2. *(Subspace Embedding:) Consider a fixed $k$-dimensional subspace $V$. If $m \geq k^2(2 + 3^q)/(\epsilon^2 \delta)$, then with probability at least $1 - \delta$, $\|xS\| = (1 \pm \epsilon)\|x\|$ simultaneously for all $x \in V$.*

---

**Algorithm 1** $k$-Space

---

1: **Input:** $A \in \mathbb{R}^{n \times d}$, $\epsilon \in (0, 1]$, integer $k$.
2: **Output:** $V \in \mathbb{R}^{n \times k}$ with orthonormal columns which spans a rank-$k$ $(1 + \epsilon)$-approximation to $\phi(A)$.

3: Set the parameters $m = \Theta(3^q k^2 + k/\epsilon)$ and $r = \Theta(3^q m^2/\epsilon^2)$.
4: Let $S$ be a $d^q \times m$ TENSORSKETCH and $T$ be an independent $d^q \times r$ TENSORSKETCH.
5: Compute $\phi(A) \cdot S$ and $\phi(A) \cdot T$.
6: Let $U$ be an orthonormal basis for the column space of $\phi(A) \cdot S$.
7: Let $W$ be the $m \times k$ matrix containing the top $k$ left singular vectors of $U^T \phi(A)T$.
8: Output $V = UW$.

---

We establish the theorem via two lemmas. The first lemma proves the approximate matrix product property via a careful second moment analysis. Due to space constraints, a proof is included only in the supplementary material version of the paper.

**Lemma 2.** *Let $A$ and $B$ be matrices with $d^q$ rows. For $m \geq (2 + 3^q)/(\epsilon^2 \delta)$, we have*

$$\Pr[\|A^T SS^T B - A^T B\|_F^2 \leq \epsilon^2 \|A\|_F^2 \|B\|_F^2] \geq 1 - \delta$$

The second lemma proves that the subspace embedding property follows from the approximate matrix product property.

**Lemma 3.** *Consider a fixed $k$-dimensional subspace $V$. If $m \geq k^2(2 + 3^q)/(\epsilon^2 \delta)$, then with probability at least $1 - \delta$, $\|xS\| = (1 \pm \epsilon)\|x\|$ simultaneously for all $x \in V$.*

*Proof.* Let $B$ be a $d^q \times k$ matrix whose columns form an orthonormal basis of $V$. Thus, we have $B^T B = I_k$ and $\|B\|_F^2 = k$. The condition that $\|xS\| = (1 \pm \epsilon)\|x\|$ simultaneously for all $x \in V$ is equivalent to the condition that the singular values of $B^T S$ are bounded by $1 \pm \epsilon$. By Lemma 2, for $m \geq (2 + 3^q)/((\epsilon/k)^2 \delta)$, with probability at least $1 - \delta$, we have

$$\|B^T SS^T B - B^T B\|_F^2 \leq (\epsilon/k)^2 \|B\|_F^4 = \epsilon^2$$

Thus, we have $\|B^T SS^T B - I_k\|_2 \leq \|B^T SS^T B - I_k\|_F \leq \epsilon$. In other words, the squared singular values of $B^T S$ are bounded by $1 \pm \epsilon$, implying that the singular values of $B^T S$ are also bounded by $1 \pm \epsilon$. Note that $\|A\|_2$ for a matrix $A$ denotes its operator norm. □

## 4 Applications

### 4.1 Approximate Kernel PCA and Low Rank Approximation

We say an $n \times k$ matrix $V$ with orthonormal columns spans a rank-$k$ $(1 + \epsilon)$-approximation of an $n \times d$ matrix $A$ if $\|A - VV^T A\|_F \leq (1 + \epsilon)\|A - A_k\|_F$. Algorithm $k$-Space (Algorithm 1) finds an $n \times k$ matrix $V$ which spans a rank-$k$ $(1 + \epsilon)$-approximation of $\phi(A)$.

Before proving the correctness of the algorithm, we start with two key lemmas. Proofs are included only in the supplementary material version of the paper.

**Lemma 4.** *Let $S \in \mathbb{R}^{d^q \times m}$ be a randomly chosen TENSORSKETCH matrix with $m = \Omega(3^q k^2 + k/\epsilon)$. Let $UU^T$ be the $n \times n$ projection matrix onto the column space of $\phi(A) \cdot S$. Then if $[U^T \phi(A)]_k$ is the best rank-$k$ approximation to matrix $U^T \phi(A)$, we have*

$$\|U[U^T \phi(A)]_k - \phi(A)\|_F \leq (1 + O(\epsilon))\|\phi(A) - [\phi(A)]_k\|_F.$$

**Lemma 5.** *Let $UU^T$ be as in Lemma 4. Let $T \in \mathbb{R}^{d^q \times r}$ be a randomly chosen TENSORSKETCH matrix with $r = O(3^q m^2/\epsilon^2)$, where $m = \Omega(3^q k^2 + k/\epsilon)$. Suppose $W$ is the $m \times k$ matrix whose columns are the top $k$ left singular vectors of $U^T \phi(A)T$. Then,*

$$\|UWW^T U^T \phi(A) - \phi(A)\|_F \leq (1 + \epsilon)\|\phi(A) - [\phi(A)]_k\|_F.$$

**Theorem 6.** *(Polynomial Kernel Rank-$k$ Space.) For the polynomial kernel of degree $q$, in $O(\text{nnz}(A)q) + n \cdot \text{poly}(3^q k/\epsilon)$ time, Algorithm $k$-SPACE finds an $n \times k$ matrix $V$ which spans a rank-$k$ $(1 + \epsilon)$-approximation of $\phi(A)$.*

*Proof.* By Lemma 4 and Lemma 5, the output $V = UW$ spans a rank-$k$ $(1 + \epsilon)$-approximation to $\phi(A)$. It only remains to argue the time complexity. The sketches $\phi(A) \cdot S$ and $\phi(A) \cdot T$ can be computed in $O(\mathtt{nnz}(A)q) + n \cdot \mathrm{poly}(3^q k/\epsilon)$ time. In $n \cdot \mathrm{poly}(3^q k/\epsilon)$ time, the matrix $U$ can be obtained from $\phi(A) \cdot S$ and the product $U^T \phi(A)T$ can be computed. Given $U^T \phi(A)T$, the matrix $W$ of top $k$ left singular vectors can be computed in $\mathrm{poly}(3^q k/\epsilon)$ time, and in $n \cdot \mathrm{poly}(3^q k/\epsilon)$ time the product $V = UW$ can be computed. Hence the overall time is $O(\mathtt{nnz}(A)q) + n \cdot \mathrm{poly}(3^q k/\epsilon)$, and the theorem follows. $\qquad\square$

We now show how to find a low rank approximation to $\phi(A)$. A proof is included in the supplementary material version of the paper.

**Theorem 7.** *(Polynomial Kernel PCA and Low Rank Factorization) For the polynomial kernel of degree $q$, in $O(\mathtt{nnz}(A)q)+(n+d)\cdot\mathrm{poly}(3^q k/\epsilon)$ time, we can find an $n\times k$ matrix $V$, a $k\times\mathrm{poly}(k/\epsilon)$ matrix $U$, and a $\mathrm{poly}(k/\epsilon)\times d$ matrix $R$ for which*

$$\|V \cdot U \cdot \phi(R) - A\|_F \le (1 + \epsilon)\|\phi(A) - [\phi(A)]_k\|_F.$$

*The success probability of the algorithm is at least .6, which can be amplified with independent repetition.*

Note that Theorem 7 implies the rowspace of $\phi(R)$ contains a $k$-dimensional subspace $L$ with $d^q \times d^q$ projection matrix $LL^T$ for which $\|\phi(A)LL^T - \phi(A)\|_F \le (1 + \epsilon)\|\phi(A) - [\phi(A)]_k\|_F$, that is, $L$ provides an approximation to the space spanned by the top $k$ principal components of $\phi(A)$.

### 4.2 Regularizing Learning With the Polynomial Kernel

Consider learning with the polynomial kernel. Even if $d \ll n$ it might be that even for low values of $q$ we have $d^q \gg n$. This makes a number of learning algorithms underdetermined, and increases the chance of overfitting. The problem is even more severe if the input matrix $A$ has a lot of redundancy in it (noisy features).

To address this, many learning algorithms add a regularizer, e.g., ridge terms. Here we propose to regularize by using rank-$k$ approximations to the matrix (where $k$ is the regularization parameter that is controlled by the user). With the tools developed in the previous subsection, this not only serves as a regularization but also as a means of accelerating the learning.

We now show that two different methods that can be regularized using this approach.

#### 4.2.1 Approximate Kernel Principal Component Regression

If $d^q > n$ the linear regression with $\phi(A)$ becomes underdetermined and exact fitting to the right hand side is possible, and in more than one way. One form of regularization is Principal Component Regression (PCR), which first uses PCA to project the data on the principal component, and then continues with linear regression in this space.

We now introduce the following approximate version of PCR.

**Definition 8.** *In the Approximate Principal Component Regression Problem (Approximate PCR), we are given an $n \times d$ matrix $A$ and an $n \times 1$ vector $b$, and the goal is to find a vector $x \in \mathbb{R}^k$ and an $n \times k$ matrix $V$ with orthonormal columns spanning a rank-$k$ $(1 + \epsilon)$-approximation to $A$ for which $x = argmin_x \|Vx - b\|_2$.*

Notice that if $A$ is a rank-$k$ matrix, then Approximate PCR coincides with ordinary least squares regression with respect to the column space of $A$. While PCR would require solving the regression problem with respect to the top $k$ singular vectors of $A$, in general finding these $k$ vectors exactly results in unstable computation, and cannot be found by an efficient linear sketch. This would occur, e.g., if the $k$-th singular value $\sigma_k$ of $A$ is very close (or equal) to $\sigma_{k+1}$. We therefore relax the definition to only require that the regression problem be solved with respect to some $k$ vectors which span a rank-$k$ $(1 + \epsilon)$-approximation to $A$.

The following is our main theorem for Approximate PCR.

**Theorem 9.** *(Polynomial Kernel Approximate PCR.) For the polynomial kernel of degree $q$, in $O(\mathtt{nnz}(A)q) + n \cdot \mathrm{poly}(3^q k/\epsilon)$ time one can solve the approximate PCR problem, namely, one*

*can output a vector $x \in \mathbb{R}^k$ and an $n \times k$ matrix $V$ with orthonormal columns spanning a rank-$k$ $(1 + \epsilon)$-approximation to $\phi(A)$, for which $x = argmin_x \|Vx - b\|_2$.*

*Proof.* Applying Theorem 6, we can find an $n \times k$ matrix $V$ with orthonormal columns spanning a rank-$k$ $(1+\epsilon)$-approximation to $\phi(A)$ in $O(\text{nnz}(A)q) + n \cdot \text{poly}(3^q k/\epsilon)$ time. At this point, one can solve solve the regression problem $argmin_x \|Vx - b\|_2$ exactly in $O(nk)$ time since the minimizer is $x = V^T b$. ☐

### 4.2.2 Approximate Kernel Canonical Correlation Analysis

In Canonical Correlation Analysis (CCA) we are given two matrices $A$, $B$ and we wish to find directions in which the spaces spanned by their columns are correlated. Due to space constraints, details appear only in the supplementary material version of the paper.

## 5 Experiments

We report two sets of experiments whose goal is to demonstrate that the $k$-Space algorithm (Algorithm 1) is useful as a feature extraction algorithm. We use standard classification and regression datasets.

In the first set of experiments, we compare ordinary $\ell_2$ regression to approximate principal component $\ell_2$ regression, where the approximate principal components are extracted using $k$-Space (we use RLSC for classification). Specifically, as explained in Section 4.2.1, we use $k$-Space to compute $V$ and then use regression on $V$ (in one dataset we also add an additional ridge regularization). To predict, we notice that $V = \phi(A) \cdot S \cdot R^{-1} \cdot W$, where $R$ is the $R$ factor of $\phi(A) \cdot S$, so $S \cdot R^{-1} \cdot W$ defines a mapping to the approximate principal components. So, to predict on a matrix $A_t$ we first compute $\phi(A_t) \cdot S \cdot R^{-1} \cdot W$ (using TENSORSKETCH to compute $\phi(A_t) \cdot S$ fast) and then multiply by the coefficients found by the regression. In all the experiments, $\phi(\cdot)$ is defined using the kernel $k(u, v) = (u^T v + 1)^3$.

While $k$-Space is efficient and gives an embedding in time that is faster than explicitly expanding the feature map, or using kernel PCA, there is still some non-negligible overhead in using it. Therefore, we also experimented with feature extraction using only a subset of the training set. Specifically, we first sample the dataset, and then use $k$-Space to compute the mapping $S \cdot R^{-1} \cdot W$. We apply this mapping to the entire dataset before doing regression.

The results are reported in Table 1. Since $k$-Space is randomized, we report the mean and standard deviation of 5 runs. For all datasets, learning with the extracted features yields better generalized errors than learning with the original features. Extracting the features using only a sample of the training set results in only slightly worse generalization errors. With regards to the MNIST dataset, we caution the reader not to compare the generalization results to the ones obtained using the polynomial kernel (as reported in the literature). In our experiments we do not use the polynomial kernel on the entire dataset, but rather use it to extract features (i.e., do principal component regularization) using only a subset of the examples (only 5,000 examples out of 60,000). One can expect worse results, but this is a more realistic strategy for very large datasets. On very large datasets it is typically unrealistic to use the polynomial kernel on the entire dataset, and approximation techniques, like the ones we suggest, are necessary.

We use a similar setup in the second set of experiments, now using linear SVM instead of regression (we run only on the classification datasets). The results are reported in Table 2. Although the gap is smaller, we see again that generally the extracted features lead to better generalization errors.

We remark that it is not our goal to show that $k$-Space is the best feature extraction algorithm of the classification algorithms we considered (RLSC and SVM), or that it is the fastest, but rather that it can be used to extract features of higher quality than the original one. In fact, in our experiments, while for a fixed number of extracted features, $k$-Space produces better features than simply using TENSORSKETCH, it is also more expensive in terms of time. If that additional time is used to do learning or prediction with TENSORSKETCH with more features, we overall get better generalization error (we do not report the results of these experiments). However, feature extraction is widely applicable, and there can be cases where having fewer high quality features is beneficial, e.g. performing multiple learning on the same data, or a very expensive learning tasks.

Table 1: Comparison of testing error with using regression with original features and with features extracted using $k$-Space. In the table, $n$ is number of training instances, $d$ is the number of features per instance and $n_t$ is the number of instances in the test set. "Regression" stands for ordinary $\ell_2$ regression. "PCA Regression" stand for approximate principal component $\ell_2$ regression. "Sample PCA Regression" stands approximate principal component $\ell_2$ regression where only $n_s$ samples from the training set are used for computing the feature extraction. In "PCA Regression" and "Sample PCA Regression" $k$ features are extracted. In $k$-Space we use $m = O(k)$ and $r = O(k)$ with the ratio between $m$ and $k$ and $r$ and $k$ as detailed in the table. For classification tasks, the percent of testing points incorrectly predicted is reported. For regression tasks, we report $\|y_p - y\|_2 / \|y\|$ where $y_p$ is the predicted values and $y$ is the ground truth.

| Dataset | Regression | PCA Regression | Sampled PCA Regression |
|---|---|---|---|
| MNIST classification $n = 60,000, d = 784$ $n_t = 10,000$ | 14% | Out of Memory | $7.9\% \pm 0.06\%$ $k = 500, n_s = 5000$ $m/k = 2$ $r/k = 4$ |
| CPU regression $n = 6,554, d = 21$ $n_t = 819$ | 12% | $4.3\% \pm 1.0\%$ $k = 200$ $m/k = 4$ $r/k = 8$ | $3.6\% \pm 0.1\%$ $k = 200, n_s = 2000$ $m/k = 4$ $r/k = 8$ |
| ADULT classification $n = 32,561, d = 123$ $n_t = 16,281$ | 15.3% | $15.2\% \pm 0.1\%$ $k = 500$ $m/k = 2$ $r/k = 4$ | $15.2\% \pm 0.03\%$ $k = 500, n_s = 5000$ $m/k = 2$ $r/k = 4$ |
| CENSUS regression $n = 18,186, d = 119$ $n_t = 2,273$ | 7.1% | $6.5\% \pm 0.2\%$ $k = 500$ $m/k = 4$ $r/k = 8$ $\lambda = 0.001$ | $6.8\% \pm 0.4\%$ $k = 500, n_s = 5000$ $m/k = 4$ $r/k = 8$ $\lambda = 0.001$ |
| USPS classification $n = 7,291, d = 256$ $n_t = 2,007$ | 13.1% | $7.0\% \pm 0.2\%$ $k = 200$ $m/k = 4$ $r/k = 8$ | $7.5\% \pm 0.3\%$ $k = 200, n_s = 2000$ $m/k = 4$ $r/k = 8$ |

Table 2: Comparison of testing error with using SVM with original features and with features extracted using $k$-Space.. In the table, $n$ is number of training instances, $d$ is the number of features per instance and $n_t$ is the number of instances in the test set. "SVM" stands for linear SVM. "PCA SVM" stand for using $k$-Space to extract features, and then using linear SVM. "Sample PCA SVM" stands for using only $n_s$ samples from the training set are used for computing the feature extraction. In "PCA SVM" and "Sample PCA SVM" $k$ features are extracted. In $k$-Space we use $m = O(k)$ and $r = O(k)$ with the ratio between $m$ and $k$ and $r$ and $k$ as detailed in the table. For classification tasks, the percent of testing points incorrectly predicted is reported.

| Dataset | SVM | PCA SVM | Sampled PCA SVM |
|---|---|---|---|
| MNIST classification $n = 60,000, d = 784$ $n_t = 10,000$ | 8.4% | Out of Memory | $6.1\% \pm 0.1\%$ $k = 500, n_s = 5000$ $m/k = 2$ $r/k = 4$ |
| ADULT classification $n = 32,561, d = 123$ $n_t = 16,281$ | 15.0% | $15.1\% \pm 0.1\%$ $k = 500$ $m/k = 2$ $r/k = 4$ | $15.2\% \pm 0.1\%$ $k = 500, n_s = 5000$ $m/k = 2$ $r/k = 4$ |
| USPS classification $n = 7,291, d = 256$ $n_t = 2,007$ | 8.3% | $7.2\% \pm 0.2\%$ $k = 200$ $m/k = 4$ $r/k = 8$ | $7.5\% \pm 0.3\%$ $k = 200, n_s = 2000$ $m/k = 4$ $r/k = 8$ |

# 6  Conclusions and Future Work

Sketching based dimensionality reduction has so far been limited to linear models. In this paper, we describe the first oblivious subspace embeddings for a non-linear kernel expansion (the polynomial kernel), opening the door for sketching based algorithms for a multitude of problems involving kernel transformations. We believe this represents a significant expansion of the capabilities of sketching based algorithms. However, the polynomial kernel has a finite-expansion, and this finiteness is quite useful in the design of the embedding, while many popular kernels induce an infinite-dimensional mapping. We propose that the next step in expanding the reach of sketching based methods for statistical learning is to design oblivious subspace embeddings for non-finite kernel expansions, e.g., the expansions induced by the Gaussian kernel.

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
