[Supplementary Material · polyker-full.pdf]

# Subspace Embeddings for the Polynomial Kernel
## (Supplementary material: the article with the omitted proofs included)

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

*Proof.* Let $C = A^T SS^T B$. We have

$$C_{u,u'} = \sum_{t=1}^{m} \sum_{i,j\in[d]^q} S(i)S(j)\delta_{i,t}\delta_{j,t} A_{i,u}B_{j,u'} = \sum_{t=1}^{m} \sum_{i\neq j\in[d]^q} S(i)S(j)\delta_{i,t}\delta_{j,t} A_{i,u}B_{j,u'} + (A^T B)_{u,u'}$$

Thus, $\mathbb{E}[C_{u,u'}] = (A^T B)_{u,u'}$.

Next, we analyze $\mathbb{E}[((C - A^T B)_{u,u'})^2]$. We have

$$((C - A^T B)_{u,u'})^2 = \sum_{t_1,t_2=1}^{m} \sum_{i_1\neq j_1, i_2 \neq j_2 \in [d]^q} S(i_1)S(i_2)S(j_1)S(j_2) \cdot \delta_{i_1,t_1}\delta_{j_1,t_1}\delta_{i_2,t_2}\delta_{j_2,t_2} \cdot$$
$$A_{i_1,u}A_{i_2,u}B_{j_1,u'}B_{j_2,u'}$$

For a term in the summation on the right hand side to have a non-zero expectation, it must be the case that $\mathbb{E}[S(i_1)S(i_2)S(j_1)S(j_2)] \neq 0$. Note that $S(i_1)S(i_2)S(j_1)S(j_2)$ is a product of random signs (possibly with multiplicities) where the random signs in different coordinates in $\{1,\ldots,q\}$ are independent and they are 4-wise independent within each coordinate. Thus, $\mathbb{E}[S(i_1)S(i_2)S(j_1)S(j_2)]$ is either 1 or 0. For the expectation to be 1, all random signs must appear with even multiplicities. In other words, in each of the $q$ coordinates, the 4 coordinates of $i_1, i_2, j_1, j_2$ must be the same number appearing 4 times or 2 distinct numbers, each appearing twice. All the subsequent claims in the proof regarding $i_1, i_2, j_1, j_2$ agreeing on some coordinates follow from this property.

Let $S_1$ be the set of coordinates where $i_1$ and $i_2$ agrees. Note that $j_1$ and $j_2$ must also agree in all coordinates in $S_1$ by the above argument. Let $S_2 \subset [q] \setminus S_1$ be the coordinates among the remaining where $i_1$ and $j_1$ agrees. Finally, let $S_3 = [q] \setminus (S_1 \cup S_2)$. All coordinates in $S_3$ of $i_1$ and $j_2$ must agree. Similarly as before, note that $i_2$ and $j_2$ agree on all coordinates in $S_2$ and $i_2$ and $j_1$ agree on all coordinates in $S_3$. We can rewrite $i_1 = (a,b,c), i_2 = (a,e,f), j_1 = (g,b,f), j_2 = (g,e,c)$ where $a, g \in [d]^{S_1}, b, e \in [d]^{S_2}, c, f \in [d]^{S_3}$.

First we show that the contribution of the terms where $i_1 = i_2$ or $i_1 = j_2$ is bounded by $\frac{2\|A_u\|^2\|B_{u'}\|^2}{m}$, where $A_u$ is the $u$th column of $A$ and $B_{u'}$ is the $u'$th column of $B$. Indeed, consider the case $i_1 = i_2$. As observed before, we must have $j_1 = j_2$ to get a non-zero contribution. Note that if $t_1 \neq t_2$, we always have $\delta_{t_1,i_1}\delta_{t_2,i_2} = 0$ as $H(i_1)$ cannot be equal to both $t_1$ and $t_2$. Thus, for fixed $i_1 = i_2, j_1 = j_2$,

$$\mathbb{E}[\sum_{t_1,t_2=1}^{m} S(i_1)S(i_2)S(j_1)S(j_2) \cdot \delta_{i_1,t_1}\delta_{j_1,t_1}\delta_{i_2,t_2}\delta_{j_2,t_2} \cdot A_{i_1,u}A_{i_2,u}B_{j_1,u'}B_{j_2,u'}]$$
$$= \mathbb{E}[\sum_{t_1=1}^{m} \delta_{i_1,t_1}^2 \delta_{j_1,t_1}^2 A_{i_1,u}^2 B_{j_1,u'}^2] = \frac{A_{i_1,u}^2 B_{j_1,u'}^2}{m}$$

Summing over all possible values of $i_1, j_1$, we get the desired bound of $\frac{\|A_u\|^2\|B_{u'}\|^2}{m}$. The case $i_1 = j_2$ is analogous.

Next we compute the contribution of the terms where $i_1 \neq i_2, j_1, j_2$ i.e. there are at least 3 distinct numbers among $i_1, i_2, j_1, j_2$. Notice that $\mathbb{E}[\delta_{i_1,t_1}\delta_{j_1,t_1}\delta_{i_2,t_2}\delta_{j_2,t_2}] \leq \frac{1}{m^3}$ because the $\delta_{i,t}$'s are 3-wise independent. For fixed $i_1, j_1, i_2, j_2$, there are $m^2$ choices of $t_1, t_2$ so the total contribution to the expectation from terms with the same $i_1, j_1, i_2, j_2$ is bounded by $m^2 \cdot \frac{1}{m^3} \cdot |A_{i_1,u}A_{i_2,u}B_{j_1,u'}B_{j_2,u'}| = \frac{1}{m}|A_{i_1,u}A_{i_2,u}B_{j_1,u'}B_{j_2,u'}|$.

Therefore,

$$\mathbb{E}[((C - A^T B)_{u,u'})^2]$$

$$\leq \frac{2\|A_u\|^2\|B_{u'}\|^2}{m} + \frac{1}{m} \sum_{\text{partition } S_1,S_2,S_3} \sum_{a,g\in[d]^{S_1},b,e\in[d]^{S_2},c,f\in[d]^{S_3}} |A_{(a,b,c),u}B_{(g,b,f),u'}A_{(a,e,f),u}B_{(g,e,c),u'}|$$

$$\leq \frac{2\|A_u\|^2\|B_{u'}\|^2}{m} + \frac{3^q}{m} \sum_{a,b,c,g,e,f} |A_{(a,b,c),u}B_{(g,b,f),u'}A_{(a,e,f),u}B_{(g,e,c),u'}|$$

$$\leq \frac{2\|A_u\|^2\|B_{u'}\|^2}{m} + \frac{3^q}{m} \sum_{g,e,f} \left(\sum_{a,b,c} A_{(a,b,c),u}^2\right)^{1/2} \left(\sum_{a,b,c} B_{(g,b,f),u'}^2 A_{(a,e,f),u}^2 B_{(g,e,c),u'}^2\right)^{1/2}$$

$$= \frac{2\|A_u\|^2\|B_{u'}\|^2}{m} + \frac{3^q\|A_u\|}{m} \sum_{g,e,f} \left(\sum_{b} B_{(g,b,f),u'}^2\right)^{1/2} \left(\sum_{a,c} A_{(a,e,f),u}^2 B_{(g,e,c),u'}^2\right)^{1/2}$$

$$\leq \frac{2\|A_u\|^2\|B_{u'}\|^2}{m} + \frac{3^q\|A_u\|}{m} \sum_{e} \left(\sum_{b,g,f} B_{(g,b,f),u'}^2\right)^{1/2} \left(\sum_{a,c,g,f} A_{(a,e,f),u}^2 B_{(g,e,c),u'}^2\right)^{1/2}$$

$$= \frac{2\|A_u\|^2\|B_{u'}\|^2}{m} + \frac{3^q\|A_u\| \cdot \|B_{u'}\|}{m} \sum_{e} \left(\sum_{a,f} A_{(a,e,f),u}^2\right)^{1/2} \left(\sum_{g,c} B_{(g,e,c),u'}^2\right)^{1/2}$$

$$\leq \frac{2\|A_u\|^2\|B_{u'}\|^2}{m} + \frac{3^q\|A_u\| \cdot \|B_{u'}\|}{m} \left(\sum_{a,e,f} A_{(a,e,f),u}^2\right)^{1/2} \left(\sum_{g,e,c} B_{(g,e,c),u'}^2\right)^{1/2}$$

$$= \frac{(2 + 3^q)\|A_u\|^2\|B_{u'}\|^2}{m},$$

where the second inequality follows from the fact that there are $3^q$ partitions of $[q]$ into 3 sets. The other inequalities are from Cauchy-Schwarz.

Combining the above bounds, we have $\mathbb{E}[((C - A^T B)_{u,u'})^2] \leq \frac{(2+3^q)\|A_u\|^2\|B_{u'}\|^2}{m}$. For $m \geq (2 + 3^q)/(\epsilon^2\delta)$, by the Markov inequality, $\|A^T S^T S B - A^T B\|_F^2 \leq \epsilon^2$ with probability $1 - \delta$.   □

The second lemma proves that the subspace embedding property follows from the approximate matrix product property.

**Lemma 3.** *Consider a fixed $k$-dimensional subspace $V$. If $m \geq k^2(2 + 3^q)/(\epsilon^2\delta)$, then with probability at least $1 - \delta$, $\|xS\| = (1 \pm \epsilon)\|x\|$ simultaneously for all $x \in V$.*

*Proof.* Let $B$ be a $d^q \times k$ matrix whose columns form an orthonormal basis of $V$. Thus, we have $B^T B = I_k$ and $\|B\|_F^2 = k$. The condition that $\|xS\| = (1 \pm \epsilon)\|x\|$ simultaneously for all $x \in V$ is equivalent to the condition that the singular values of $B^T S$ are bounded by $1 \pm \epsilon$. By Lemma 2, for $m \geq (2 + 3^q)/((\epsilon/k)^2\delta)$, with probability at least $1 - \delta$, we have

$$\|B^T S S^T B - B^T B\|_F^2 \leq (\epsilon/k)^2\|B\|_F^4 = \epsilon^2$$

Thus, we have $\|B^T S S^T B - I_k\|_2 \leq \|B^T S S^T B - I_k\|_F \leq \epsilon$. In other words, the squared singular values of $B^T S$ are bounded by $1 \pm \epsilon$, implying that the singular values of $B^T S$ are also bounded by $1 \pm \epsilon$. Note that $\|A\|_2$ for a matrix $A$ denotes its operator norm.   □

# 4   Applications

## 4.1   Approximate Kernel PCA and Low Rank Approximation

We say an $n \times k$ matrix $V$ with orthonormal columns spans a rank-$k$ $(1 + \epsilon)$-approximation of an $n \times d$ matrix $A$ if $\|A - VV^T A\|_F \leq (1 + \epsilon)\|A - A_k\|_F$. Algorithm $k$-Space (Algorithm 1) finds an $n \times k$ matrix $V$ which spans a rank-$k$ $(1 + \epsilon)$-approximation of $\phi(A)$.

Before proving the correctness of the algorithm, we start with two key lemmas.

**Algorithm 1** $k$-Space

---

1: **Input:** $A \in \mathbb{R}^{n \times d}$, $\epsilon \in (0, 1]$, integer $k$.
2: **Output:** $V \in \mathbb{R}^{n \times k}$ with orthonormal columns which spans a rank-$k$ $(1 + \epsilon)$-approximation to $\phi(A)$.

3: Set the parameters $m = \Theta(3^q k^2 + k/\epsilon)$ and $r = \Theta(3^q m^2/\epsilon^2)$.
4: Let $S$ be a $d^q \times m$ TENSORSKETCH and $T$ be an independent $d^q \times r$ TENSORSKETCH.
5: Compute $\phi(A) \cdot S$ and $\phi(A) \cdot T$.
6: Let $U$ be an orthonormal basis for the column space of $\phi(A) \cdot S$.
7: Let $W$ be the $m \times k$ matrix containing the top $k$ left singular vectors of $U^T \phi(A)T$.
8: Output $V = UW$.

---

**Lemma 4.** *Let $S \in \mathbb{R}^{d^q \times m}$ be a randomly chosen TENSORSKETCH matrix with $m = \Omega(3^q k^2 + k/\epsilon)$. Let $UU^T$ be the $n \times n$ projection matrix onto the column space of $\phi(A) \cdot S$. Then if $[U^T \phi(A)]_k$ is the best rank-$k$ approximation to matrix $U^T \phi(A)$, we have*

$$\|U[U^T \phi(A)]_k - \phi(A)\|_F \leq (1 + O(\epsilon))\|\phi(A) - [\phi(A)]_k\|_F.$$

*Proof.* The proof is the same as Theorem 4.2 of Clarkson and Woodruff [4], which is in turn based on Theorem 3.2 of the same work. While that theorem is stated for a different family of sketching matrices $S$, the only properties used about $S$ in the proof are that with constant probability:

1. (Subspace Embedding:) For any $k \times d^q$ matrix $V^T$ with orthonormal rows, for all $z \in \mathbb{R}^n$, $\|zV^T S\|_2 = (1 \pm \epsilon_0)\|zV^T\|_2$, where $\epsilon_0$ is a sufficiently small constant. That is, $S$ is a subspace embedding for the rowspace of $V^T$, and

2. (Approximate Matrix Product:) For any two matrices $A, B^T$ with $d^q$ columns, $\|ASS^T B - AB\|_F \leq \sqrt{\frac{\epsilon}{k}} \cdot \|A\|_F \|B\|_F$.

For $S$ being a TENSORSKETCH and our choice of $m$, the first property follows from Lemma 3, while the second property follows from Lemma 2. Applying the conclusion of the said Theorem 4.2, we have that the column space of $\phi(A)S$ contains a subspace $V$ of dimension $k$ which $V$ spans a rank-$k$ $(1 + \epsilon)$-approximation of $\phi(A)$. Finally, applying Lemma 4.3 of Clarkson and Woodruff [4], if $UU^T$ is the $n \times n$ projection matrix onto the column space of $\phi(A)S$, then

$$\|U[U^T \phi(A)]_k - \phi(A)\|_F \leq (1 + O(\epsilon))\|\phi(A) - [\phi(A)]_k\|_F,$$

as required. $\qquad\qquad\qquad\qquad\qquad\qquad\qquad\qquad\qquad\qquad\qquad\qquad\qquad\qquad\qquad\square$

**Lemma 5.** *Let $UU^T$ be as in Lemma 4. Let $T \in \mathbb{R}^{d^q \times r}$ be a randomly chosen TENSORSKETCH matrix with $r = O(3^q m^2/\epsilon^2)$, where $m = \Omega(3^q k^2 + k/\epsilon)$. Suppose $W$ is the $m \times k$ matrix whose columns are the top $k$ left singular vectors of $U^T \phi(A)T$. Then,*

$$\|UWW^T U^T \phi(A) - \phi(A)\|_F \leq (1 + \epsilon)\|\phi(A) - [\phi(A)]_k\|_F.$$

*Proof.* The proof is implicit in the proof of Theorem 1.5 of Kannan et al. [9]. Namely, Theorem 4.1 of Kannan et al. [9] applied to the matrix $U^T \phi(A)$ implies that $\|WW^T U^T \phi(A) - U^T \phi(A)\|_F^2 \leq (1 + O(\epsilon))\|[U^T \phi(A)]_k - U^T \phi(A)\|_F^2$. This then implies, using the same derivation of equations (2), (3), (4), (5), (6), and (7) of the proof of Theorem 1.5 of [9] that $\|UWW^T U^T \phi(A) - \phi(A)\|_F \leq (1 + \epsilon)\|\phi(A) - [\phi(A)]_k\|_F$. $\qquad\square$

**Theorem 6.** *(Polynomial Kernel Rank-$k$ Space.) For the polynomial kernel of degree $q$, in $O(\text{nnz}(A)q) + n \cdot \text{poly}(3^q k/\epsilon)$ time, Algorithm $k$-SPACE finds an $n \times k$ matrix $V$ which spans a rank-$k$ $(1 + \epsilon)$-approximation of $\phi(A)$.*

*Proof.* By Lemma 4 and Lemma 5, the output $V = UW$ spans a rank-$k$ $(1 + \epsilon)$-approximation to $\phi(A)$. It only remains to argue the time complexity. The sketches $\phi(A) \cdot S$ and $\phi(A) \cdot T$ can be computed in $O(\text{nnz}(A)q) + n \cdot \text{poly}(3^q k/\epsilon)$ time. In $n \cdot \text{poly}(3^q k/\epsilon)$ time, the matrix $U$ can be obtained from $\phi(A) \cdot S$ and the product $U^T \phi(A)T$ can be computed. Given $U^T \phi(A)T$, the matrix

$W$ of top $k$ left singular vectors can be computed in $\mathrm{poly}(3^q k/\epsilon)$ time, and in $n \cdot \mathrm{poly}(3^q k/\epsilon)$ time the product $V = UW$ can be computed. Hence the overall time is $O(\mathrm{nnz}(A)q) + n \cdot \mathrm{poly}(3^q k/\epsilon)$, and the theorem follows. $\qquad\square$

We now show how to find a low rank approximation to $\phi(A)$.

**Theorem 7.** *(Polynomial Kernel PCA and Low Rank Factorization) For the polynomial kernel of degree $q$, in $O(\mathrm{nnz}(A)q) + (n+d) \cdot \mathrm{poly}(3^q k/\epsilon)$ time, we can find an $n \times k$ matrix $V$, a $k \times \mathrm{poly}(k/\epsilon)$ matrix $U$, and a $\mathrm{poly}(k/\epsilon) \times d$ matrix $R$ for which*

$$\|V \cdot U \cdot \phi(R) - A\|_F \le (1+\epsilon)\|\phi(A) - [\phi(A)]_k\|_F.$$

*The success probability of the algorithm is at least .6, which can be amplified with independent repetition.*

*Proof.* By Theorem 6, in $O(\mathrm{nnz}(A)q) + n\,\mathrm{poly}(3^q k/\epsilon)$ time we can find an $n \times k$ matrix $V$ which spans a rank-$k$ $(1+\epsilon)$-approximation of $\phi(A)$ with probability at least .9. We now apply a sampling theorem of Drineas, Mahoney, and Muthukrishnan:

**Fact 8.** *(Theorem 5 of [6], restated) Suppose $V$ is an $n \times k$ matrix with orthonormal columns, $B \in \mathbb{R}^{n \times m}$, and $\epsilon \in (0,1]$. Define sampling probabilities $p_i = \frac{1}{k}\|V_i\|_2^2$, where $V_i$ is the $i$-th row of $V$ and $i \in [n]$. Consider the following algorithm:*

1. *Randomly sample a set of $3200k^2/\epsilon^2$ rows of $V$. Create an $n \times n$ diagonal matrix $S$ for which $S_{i,i} = 0$ if $V_i$ is not sampled, and $S_{i,i} = 1/\sqrt{p_i}$ if $V_i$ is sampled.*

2. *Output $\tilde{X} = (SV)^+ SB$.*

*Then with probability at least .7,*

$$\|B - V\tilde{X}\|_F \le (1+\epsilon) \min_{X \in \mathbb{R}^{k \times m}} \|B - VX\|_F.$$

We apply Fact 8 with our matrix $V$ which spans a rank-$k$ $(1+\epsilon)$-approximation of $\phi(A)$ and with the matrix $B$ equal to $\phi(A)$. With these parameters Fact 8 and a union bound implies that with probaility at least .6,

$$\|\phi(A) - V((SV)^+ S)\phi(A)\|_F \le$$
$$(1+\epsilon) \min_{X \in \mathbb{R}^{k \times m}} \|\phi(A) - VX\|_F \le$$
$$(1+\epsilon)^2 \|\phi(A) - [\phi(A)]_k\|_F \,,$$

and so in our low rank decomposition $V \cdot U \cdot \phi(R)$ we set $U = (SV)^+ S$ and $R$ to be the subset of $3200k^2/\epsilon^2$ rows of $A$ that are sampled by the algorithm in $\phi(A)$. As $(1+\epsilon)^2 = 1 + O(\epsilon)$, the correctness guarantee follows by rescaling $\epsilon$ by a constant factor.

We can perform the sampling in $O(nk)$ time, from which we can form the matrix $S$ and the matrix $R$. The total time is $O(\mathrm{nnz}(A)q) + (n+d) \cdot \mathrm{poly}(3^q k/\epsilon)$. $\qquad\square$

Note that Theorem 7 implies the rowspace of $\phi(R)$ contains a $k$-dimensional subspace $L$ with $d^q \times d^q$ projection matrix $LL^T$ for which $\|\phi(A)LL^T - \phi(A)\|_F \le (1+\epsilon)\|\phi(A) - [\phi(A)]_k\|_F$, that is, $L$ provides an approximation to the space spanned by the top $k$ principal components of $\phi(A)$.

## 4.2 Regularizing Learning With the Polynomial Kernel

Consider learning with the polynomial kernel. Even if $d \ll n$ it might be that even for low values of $q$ we have $d^q \gg n$. This makes a number of learning algorithms underdetermined, and increases the chance of overfitting. The problem is even more severe if the input matrix $A$ has a lot of redundancy in it (noisy features).

To address this, many learning algorithms add a regularizer, e.g., ridge terms. Here we propose to regularize by using rank-$k$ approximations to the matrix (where $k$ is the regularization parameter

that is controlled by the user). With the tools developed in the previous subsection, this not only serves as a regularization but also as a means of accelerating the learning.

We now show that two different methods that can be regularized using this approach.

### 4.2.1 Approximate Kernel Principal Component Regression

If $d^q > n$ the linear regression with $\phi(A)$ becomes underdetermined and exact fitting to the right hand side is possible, and in more than one way. One form of regularization is Principal Component Regression (PCR), which first uses PCA to project the data on the principal component, and then continues with linear regression in this space.

We now introduce the following approximate version of PCR.

**Definition 9.** *In the Approximate Principal Component Regression Problem (Approximate PCR), we are given an $n \times d$ matrix $A$ and an $n \times 1$ vector $b$, and the goal is to find a vector $x \in \mathbb{R}^k$ and an $n \times k$ matrix $V$ with orthonormal columns spanning a rank-$k$ $(1 + \epsilon)$-approximation to $A$ for which $x = argmin_x \|Vx - b\|_2$.*

Notice that if $A$ is a rank-$k$ matrix, then Approximate PCR coincides with ordinary least squares regression with respect to the column space of $A$. While PCR would require solving the regression problem with respect to the top $k$ singular vectors of $A$, in general finding these $k$ vectors exactly results in unstable computation, and cannot be found by an efficient linear sketch. This would occur, e.g., if the $k$-th singular value $\sigma_k$ of $A$ is very close (or equal) to $\sigma_{k+1}$. We therefore relax the definition to only require that the regression problem be solved with respect to some $k$ vectors which span a rank-$k$ $(1 + \epsilon)$-approximation to $A$.

The following is our main theorem for Approximate PCR.

**Theorem 10.** *(Polynomial Kernel Approximate PCR.) For the polynomial kernel of degree $q$, in $O(\texttt{nnz}(A)q) + n \cdot \text{poly}(3^q k/\epsilon)$ time one can solve the approximate PCR problem, namely, one can output a vector $x \in \mathbb{R}^k$ and an $n \times k$ matrix $V$ with orthonormal columns spanning a rank-$k$ $(1 + \epsilon)$-approximation to $\phi(A)$, for which $x = argmin_x \|Vx - b\|_2$.*

*Proof.* Applying Theorem 6, we can find an $n \times k$ matrix $V$ with orthonormal columns spanning a rank-$k$ $(1 + \epsilon)$-approximation to $\phi(A)$ in $O(\texttt{nnz}(A)q) + n \cdot \text{poly}(3^q k/\epsilon)$ time. At this point, one can solve solve the regression problem $argmin_x \|Vx - b\|_2$ exactly in $O(nk)$ time since the minimizer is $x = V^T b$. □

### 4.2.2 Approximate Kernel Canonical Correlation Analysis

In Canonical Correlation Analysis (CCA) we are given two matrices $A$, $B$ and we wish to find directions in which the spaces spanned by their columns are correlated. A formal linear algebraic definition of CCA is as follows.

**Definition 11.** *Let $A \in \mathbb{R}^{m \times n}$ and $B \in \mathbb{R}^{m \times \ell}$, and assume that $p = \text{rank}(A) \geq \text{rank}(B) = q$. The* canonical correlations $\sigma_1(A, B) \geq \sigma_2(A, B) \geq \cdots \geq \sigma_q(A, B)$ *of the matrix pair $(A, B)$ are defined recursively by the following formula:*

$$\sigma_i(A, B) = \max_{x \in \mathcal{A}_i, y \in \mathcal{B}_i} \sigma(Ax, By) =: \sigma(Ax_i, By_i),$$

$$i = 1, \ldots, q$$

*where*

- $\sigma(u, v) = |u^T v| / (\|u\|_2 \|v\|_2)$,

- $\mathcal{A}_i = \{x : Ax \neq \mathbf{0}, \mathbf{Ax} \perp \{\mathbf{Ax_1}, \ldots, \mathbf{Ax_{i-1}}\}\}$,

- $\mathcal{B}_i = \{y : By \neq \mathbf{0}, \mathbf{By} \perp \{\mathbf{By_1}, \ldots, \mathbf{By_{i-1}}\}\}$.

*The unit vectors $Ax_1/\|Ax_1\|_2, \ldots, Ax_q/\|Ax_q\|_2$, $By_1/\|By_1\|_2, \ldots, By_q/\|By_q\|_2$ are called the* canonical *or* principal *vectors. The vectors $x_1/\|Ax_1\|_2, \ldots, x_q/\|Ax_q\|_2$, $y_1/\|By_1\|_2, \ldots, y_q/\|By_q\|_2$ are called* canonical weights *(or* projection *vectors). Note that the canonical weights and the canonical vectors are* not *uniquely defined.*

CCA finds correlations between any parts of the spectrum. Potentially, random correlation between noise might be found, skewing the results. The problem is aggravated in the kernel setting (where we find correlations between $\phi(A)$ and $\phi(B)$): if $d^q > n$ then there is exact correlation between the subspaces. One common way to address this is to add ridge terms.

We consider a different form of regularization: finding the correlations between the dominant subspaces of $A$ and $B$ (their principal components). We now introduce an approximate version of it:

**Definition 12.** *In the Approximate Principal Component CCA Problem (Approximate PC-CCA), we are given an $n \times d_1$ matrix $A$ and an $n \times d_2$ matrix $B$, and the goal is to find two $n \times k$ orthonormal matrices $U$ and $V$, where $U$ spans a rank-k approximation to $A$ and $V$ spans a rank-k approximation to $B$, and output the CCA between $U$ and $V$.*

**Theorem 13.** *(Polynomial Kernel Approximate PCR.) For the polynomial kernel of degree q, in $O((\mathtt{nnz}(A) + \mathtt{nnz}(B))q) + n \cdot \mathrm{poly}(3^q k/\epsilon)$ time one can solve the approximate PC-CCA problem on $\phi(A)$ and $\phi(B)$.*

*Proof.* Applying Theorem 6, we can find an $n \times k$ matrix $V$ with orthonormal columns spanning a rank-$k$ $(1 + \epsilon)$-approximation to $\phi(A)$ in $O(\mathtt{nnz}(A)q) + n \cdot \mathrm{poly}(3^q k/\epsilon)$ time, a $n \times k$ matrix $U$ with orthonormal columns spanning a rank-$k$ $(1 + \epsilon)$-approximation to $\phi(B)$ in $O(\mathtt{nnz}(B)q) + n \cdot \mathrm{poly}(3^q k/\epsilon)$ time. At this point, we simply compute the CCA between $U$ and $V$, which amounts to computing an SVD on $U^T V$. This takes $O(nk^2)$ time. $\qquad\square$