[Reviews · NeurIPS 2014]

Submitted by Assigned_Reviewer_12

Summary: The authors provide a novel method for oblivious subspace embedding that is computationally efficient in a sense that it circumvents explicit computation of feature map. Essentially, what the authors suggest is a special way to do implicit low-rank approximation of feature matrix. Asymptotic properties of that approximation are also shown in the paper. The paper mostly focuses on feature map with polynomial kernel, leaving out infinite dimensional kernels for further research.

Quality, Clarity: This is a high quality paper with detailed technical proofs. Clarity is on the high level as well: both proofs and examples of applications are provided. There is a strong significance as well since it shows a way to approximate feature matrix without its explicit computation, which boost kernel-based algorithms computationally.

Originality, significance: method proposed in paper is considerable improvement to existing algorithms, however the major contribution here is empirical. In that sense, assumption of polynomial (in general - finite dimensional) kernel is too restrictive. A harder challenge in computing feature map happens when kernel space is infinite dimensional (e.g. gaussian kernel), so it would be a greater and more useful contribution if authors show how to implicitly approximate feature map for those types of kernels. In vast amounts of literature (e.g. on learning kernels) people are interested what to do if user is given a finite set of arbitrary kernels, e.g. how to combine them for your learning algo (which can be both supervised and unsupervised). Thus, extending the author's subspace embedding method to arbitrary kernel, which could be infinite dimensional as well could be a great generalization.
Summary: Pros: Makes oblivious learning faster and more achievable computationally. Asymptotic properties of subspace embedding also hold.

Cons: Restriction to polynomial kernel decreases the significance of this paper.

Submitted by Assigned_Reviewer_27

This paper proposed a sketching based method for approximating polynomial kernels.
What is interesting about the paper is that the method approximately preserves the principal subspace of the kernel feature space (this property is called oblivious subspace embedding property in the paper). Furthermore, the method can work with the data directly without the need to first obtain the square kernel matrix.

The sketching method in the paper, TensorSketch, has been proposed by Pagh [3]. The key result of the paper is proving the oblivious subspace embedding property of TensorSketch for polynomial kernels. The authors also applied the approximated kernel for downstream learning tasks such as kernel PCA and kernel PCA regression.

The derivation of the results is rigorous and seems correct to me. The paper is also organized clearly, but the description of CountSketch and TensorSketch are highly compressed and technical. No enough intuitive and concrete explanation why it is particularly good for this polynomial features. This also makes the reading of the proof of Lemma 2 more difficult.

This experiments is not confirming anything developed in the theory section. It is showing aspect related to generalization ability of downstream algorithms. The key selling points of the paper, efficiency and oblivious subspace embedding, are not experimented with. Many other questions arise from the experiments. For instance, have the kernel regression and svm on mnist dataset been properly regularized? Using polynomial kernels, one can achieve a few percent error in mnist (http://yann.lecun.com/exdb/mnist/).
Summary: This paper proposed a sketching based method for approximating polynomial kernels.
It is interesting that the method approximately preserves the principal subspace of the kernel feature space. It is not clear what the experiments are showing.

Submitted by Assigned_Reviewer_31

The paper provides an algorithm to find a subspace embedding for polynomial kernel.

The authors claim to propose the first fast OSE for non-linear kernel. This statement appears confusing to me. First of all, there are works on approximating OSE for non-linear kernel, for example [1] below, which does not mention in the paper at all. And the algorithm in [1] could be used for Gaussian kernel. Thus I would suppose that this paper focuses on the efficiency of the algorithm.

Regarding the main theorem, it does provide a clear explanation that tensor sketch, in terms of \phi(v) T could be understood as an OSE. I do not manage to read the whole proof, but the intuition seems to be right. One of the main issues I have in mind is about the 5. in the Algorithm 1. The authors mention that \phi(v) T could be computed efficiently by [3]. However, it is not so intuitive for me since the original paper is not about Tensor Sketch directly. I suggest the authors provides more details on this, at least in the supplementary materials, since this is an important part of the algorithms. Regarding the efficiency, it would be interesting it the authors provide some comparisons between other methods for OSE.

In the experiment section, the authors make some contradicting statement with the claim about efficiency of the algorithm. For example, "K-space can be rather expensive both in time and in memory to compute". This really confuses me very much, since as I understand one important point about the algorithm is that it gives a subspace embedding in polynomial time. Moreover, the experiments shows that the features extracted by the algorithm improves the performance of supervised learning compared to using the original features. This result makes sense, but it is already well-known from previous works that OSE usually gives better performances than original features. One interesting experiment I want to see would be the comparison between the proposed algorithm and the existing OSE algorithms.

Please also be more careful about the mathematical formula in the paper. For example, at line 302 I should be "V U \phi(R) - \phi(A)" if I understand correctly.

[1] Kernelized Locality-Sensitive Hashing
Summary: The paper proposes an algorithm for subspace embedding of polynomial kernels, which is demonstrated very clear theoretically. However, the experiments do not reflect the efficiency of the algorithm very well, which I consider as one of the main contributions of the paper and lack comparisons with other OSE algorithms.
Author Feedback
Author rebuttal: We thank the reviewers for their review, and for their helpful suggestions for improving the paper. We will incorporate the proposals in the final version, including more details and intuition, as well as correct some typos found by the reviewers. In the following we answer the main issues raised by the reviewers:

1) While the paper “Kernelized Locality-Sensitive Hashing for Scalable Image Search” does describe a procedure for obtaining an embedding (much like PCA), it is not an OSE. It is unfortunate that in the paper we did not elaborate what we mean by “OSE”, beyond a sentence saying “An OSE is a data-independent random transform which is an approximate isometry over a subspace”. However, that sentence is quite telling; an OSE has two components:

- It is an approximate isometry. That is condition 2 in Theorem 1.

- It is data independent.

These conditions are crucial for the theoretical guarantees in the paper. For Kernelized LSH, it is not data-independent, and there is no proof that it is an approximate isometry. While there are some methods for non-linear kernels that generate an approximate isometry (e.g. Kernel PCA), or methods that are data independent (like the Random Fourier Features of Rahimi and Recht), previously there were no known OSEs for non-linear kernels.

Clearly explaining what is an OSE and why it is important is an unfortunate omission on our part, which we will correct in the final version of the paper. We want to note that there are no other OSEs we are aware of that can be applied without first obtaining the square kernel matrix or expanding all the polynomial features; even performing this first step before applying a different kind of OSE would be considerably slower than what we are proposing.

2) Regarding the results for MNIST: we note that we do not use the full polynomial kernel (with rank regularization or other regularization) but only extract features (i.e., do principal component regularization) using only a subset of the examples (only 5000 examples out of 60,000). One can expect worse results than one can get by using the full polynomial kernel on the entire data, but this situation is more representative for very large datasets, where it is unrealistic to use the polynomial kernel on the entire data.

It is also the case that we did not explore the best setting for regularization, and degree of the kernel, or the constant parameter, etc., since the goal was mainly to show that the proposed k-space algorithm can be successfully used to extract features and not to propose the best possible algorithm for MNIST.

3) The statement “K-space can be rather expensive both in time and in memory to compute” was badly phrased. The algorithm is efficient and gives an embedding in time that is faster than explicitly expanding the feature map, or using kernel PCA. However, there is still some non-negligible overhead in using it, and one can hope to mitigate that by using the k-space algorithm on only a subset of the examples, and then using the embedding on the entire dataset. In the paper we explore this scenario, and show that typically there is little penalty in doing so.

(4) Regarding step 5 in Algorithm 1 we can add more detail. The idea is that each row of phi(A) is the tensor product of q copies of the corresponding row of A, and phi(A)*S means applying S to each of the rows of phi(A). To do so, as outlined in section 2, for each row A_i we create q polynomials p_1(x), ..., p_q(x) as described there in O(nnz(A_i)q) time. To compute the product of these q polynomials mod x^B-1, we multiply p_1(x) * p_2(x) using the algorithm for multiplying two polynomials in O(B log B) time given by Pagh based on the FFT, obtaining a new polynomial r(x) which may have degree as large as 2B-2. We then in O(B) time reduce the polynomial r to degree at most B-1 using the rule x^B = 1. We then multiply r(x) * p_3(x) using the same trick, etc., and in total we obtain O(q B log B) time for computing the product of the q polynomials mod x^B-1. We then apply this to each of the rows of phi(A) in turn. In Algorithm 1, phi(A) has n rows, and the parameter B from the introduction corresponds to m in the computation of phi(A)*S, so the time to compute phi(A)*S is O(nnz(A)q) + O(n q m log m) = O(nnz(A)q) + n*poly(3^q k/eps).